# Retrieving Landmark Salience Based on Wikipedia: An Integrated Ranking Model

## Noa Binski [1], Asya Natapov [2,*] and Sagi Dalyot [1]

1    Mapping and Geo-Information Engineering, Civil and Environmental Engineering Faculty, Technion – Israel Institute of Technology, Technion City, Haifa 3200003, Israel; sbinski@campus.technion.ac.il (N.B.); dalyot@technion.ac.il (S.D.)
2    Centre for Advanced Spatial Analysis, Bartlett Faculty of the Built Environment, University College London, London WC1E 6BT, UK
*    Correspondence: a.natapov@ucl.ac.uk; Tel.: +020-3108-3876

**Abstract:** Landmarks are important for assisting in wayfinding and navigation and for enriching user experience. Although many user-generated geotagged sources exist, landmark entities are still mostly retrieved from authoritative geographic sources. Wikipedia, the world's largest free encyclopedia, stores geotagged information on many geospatial entities, including a very large and well-founded volume of landmark information. However, not all Wikipedia geotagged landmark entities can be considered valuable and instructive. This research introduces an integrated ranking model for mining landmarks from Wikipedia predicated on estimating and weighting their salience. Other than location, the model is based on the entries' category and attributed data. Preliminary ranking is formulated on the basis of three spatial descriptors associated with landmark salience, namely permanence, visibility, and uniqueness. This ranking is integrated with a score derived from a set of numerical attributes that are associated with public interest in the Wikipedia page—including the number of redirects and the date of the latest edit. The methodology is comparatively evaluated for various areas in different cities. Results show that the developed integrated ranking model is robust in identifying landmark salience, paving the way for incorporation of Wikipedia's content into navigation systems.

**Keywords:** Wikipedia; landmark salience; spatial cognition; data mining; user-generated content

## 1. Introduction

### 1.1. Landmarks and Human Spatial Cognition

The way we navigate has significantly changed in the last few decades, and it is hard to imagine navigation today without using a navigation system. Route directions used by navigation systems give a relatively small and simple set of guidelines based on turn by turn instructions. However, our brains perceive geography cognitively in the form of distinctive landmarks and shared experiences. While a navigation system will tell you: "in 500 meters turn left", a person will guide you: "take the first left behind the big tower." More specifically, humans create a mental (cognitive) map that lies at the basis of route and survey knowledge, spatial awareness, and environmental perception [1,2]. These mental maps rely on a number of neural mechanisms associated with different brain regions [3,4]. At the heart of this system are 'place cells' in the hippocampus (an area in our brain that is involved in the formation of new memories, learning, and emotions). 'Place cells' are neurons that encode spatial information, such as the relative location in relation to landmarks, and the connections between the current location and other locations [5,6]. Importantly, these neural mechanisms rely on the perception and saliency of

landmarks, their relevance, importance and features. Therefore, landmarks provide an essential basis for navigation [7–9], and help in better understanding, and visualizing the environment [10]. A wide body of research shows that landmarks have the capacity to dramatically improve the user experience and wayfinding effectiveness [11–19].

### 1.2. Landmarks and Human Spatial Cognition Landmark Definition

The word landmark is a label for several different concepts that are often vaguely specified [20]. The Urban Knowledge Data Structure elaborates what types of geographic features may serve as contributing landmarks: " . . . from signage found along a street and individual buildings, such as churches, to linear features, such as rivers, to salient street intersections, such as roundabouts" [21]. Among the different meanings of the term "landmark", this study refers to a recognizable natural or man-made feature; a feature that stands out from its surroundings. The term is also applied to smaller structures or features that have a collective meaning and become local or national symbols [22].

There are two common categories of landmarks, global and local. Global landmarks are distinguishable from a distance, while local landmarks are on-route landmarks [23]. In this paper we conduct landmark retrieval and estimate landmark salience, so its scope is limited to global landmarks.

### 1.3. Advantages of User-Generated Content for Landmark Mining

Initially, authoritative and commercial sources, such as topographic and navigation data maps (e.g., Here.com and TomTom), were used to identify and to extract landmarks. However, the increasing availability and popularity of User-Generated Content (UGC), e.g., geotagged locations and information shared on online social media, gradually turn it into a valuable alternative [24]. Data and information retrieved from UGC can serve as an enriched source for the retrieval of more informal information, such as vernacular places and local knowledge, which is heavily used by people in their daily lives, but not represented in authoritative digital maps [25]. Online UGC, such as the collection of geotagged photos on social media, e.g., Flickr, can be used to measure the popularity of locations and to infer users' preferences of landmarks for travel suggestions [26–28]. OpenStreetMap can be used for data extraction, weighing, and selection of landmarks, as well as the generation of landmark-based navigation instructions for pedestrian routes [29]. In [30] the same data source is used to estimate visual characteristics of the potential landmarks coupled with geometric calculations about the route. A landmark index based on 3D city models and OSM is developed in [31]. This index is assigned to every building to assess the suitability of a building as a landmark.

As Quesnot et al. [32] argues, the social dimension (i.e., the way an object is recognized by a person or a group of people) represents an important component in our perception of place, but still often is excluded from landmark mining systems. Wikipedia, the UGC largest encyclopedia, is an enormous body of expertise that contains vernacular and local knowledge and reflects public interests across numerous dimensions. This study proposes an innovative method to use Wikipedia's potential to enrich existing sources, as it offers cost-effective information of geospatial entities [24,25,33].

### 1.4. The Existing Knowledge Gap

The inclusion of landmarks into navigation systems is a long-standing goal. First, the system has to be able to extract suitable points of interest and to assess their salience in the role of potential landmarks. Then, they have to be integrated in meaningful ways adjusting to the particular travel mode, navigation context or appropriate arrangements. In this paper we focus only on the first stage, conducting landmark retrieval and estimating landmark salience. This stage is paving the way for the next one that requires an alternative set of preconditions to define an appropriate landmark for a particular wayfinding task.

So far, the complete integration of landmark-based navigation has fallen short due to significant difficulties to obtain a sufficient data source. To close this gap, we investigate the use of Wikipedia for identifying and retrieving landmark information.

At this point we focus on data mining alone, and not on the enrichment of the navigation systems. As far as we know, it is the first attempt to use Wikipedia, which is not a geographic data source by nature, to construct a formulation enabling the retrieval of landmarks' salience.

However, a straightforward information retrieval based solely on the geotagged (location) attribution will not suffice. Such trivial data retrieval will yield redundant results, including many inessential and meaningless landmarks. To appraise landmarks' salience, and to weight them accordingly, we introduce a unique integrated ranking model. This model is based on quantitative and qualitative metrics and rules used for the hierarchical classification of Wikipedia entries. It is designed to allow the identification and retrieval of the most valuable and prominent landmarks, filtering redundant and less significant ones. The classification itself does not rely solely on the location and the Wikipedia category of the landmark, but incorporates three spatial descriptors, namely permanence, visibility, and uniqueness, which should attest the landmark's salience. These are integrated with a score that is retrieved from a formulation of numerical attributes. The attributes do not have a spatial context but are associated with the popularity of the Wikipedia page, namely the page statistics. These statistics point to the public interest in the Wikipedia entry as being significant and distinctive, thus the ranking incorporates a crowdsourced context.

The aim of the proposed ranking model is to investigate how Wikipedia can contribute to the estimation of landmark salience, which in turn should enrich the overall user experience and effectiveness of various Location Based Services (LBSs). Using several experimental examples, we show the potential the model holds for mining UGC large databases for Geographic Information Retrieval (GIR) and knowledge recovery of valuable environmental information.

The remainder of the paper is organized as follows: Section 2 describes related work; Section 3 presents the overall architecture of our proposed ranking framework and methods; Section 4 evaluates the integrated ranking model by several comparative trials; and finally, Section 5 concludes the paper with a discussion, and describes further research directions.

## 2. Related Research

### 2.1. Data Mining of Wikipedia

Wikipedia entries are stylistically indistinguishable from traditional printed sources (i.e., encyclopedias), since it is based on crowdsourcing, having many authors and revision cycles [34]. A Wikipedia template, e.g., "Media Wiki Templates", has an inherent structure that can be used to extract consistent and meaningful information [35]. These templates are also used in DBpedia, a community-based project, which retrieves structured information from Wikipedia, allowing one to perform sophisticated queries and text mining procedures, freely available on the web [26]. Overell et al. [36] showed how to disambiguate place names mentioned in Wikipedia to locations in gazetteers, where the authors developed geographically based disambiguation methods, instead of using semantic similarities. Hoffart et al. [37] presented YAGO2 knowledge base, in which entities, facts, and events from Wikipedia (among others) are anchored to time and space, while relying on the integration of the spatio-temporal dimensions. GIR, in general, involves the retrieval of documents based on the relevance of their geographic and thematic attribution and content. As concluded by Clough et al. [38] and Medelyan et al. [39], the spatial relevance should be considered independently from the thematic relevance. Santos et al. [40] introduced the GikiP tool aimed at retrieving geographically-related information from Wikipedia based on a combination of methods from GIR and question answering. Although authors showed that the tool produced reliable results, dealing with issues of redundancy and non-relevant information, among others, remained open. Ahlers [41] discussed how GIR from UGC is still limited in terms of informal address schemes or landmark-oriented location references, suggesting that geoparsing and geocoding of Wikipedia entries could be found suitable.

Gensel and Tomko [42] integrated users into the landmark selection by introducing a mobile application that enables a user-generated collection of landmarks. They used Wikipedia to determine the cultural and historical significance of the collected landmarks. In Popescu and Grefenstette [43] Wikipedia was used to create personalized recommendations for tourists by mining and extracting landmarks that might serve as touristic destinations. Authors exploited a vocabulary of feature-specific elements (such as palace, skyscraper, park, or museum) and argued that Wikipedia stores qualitative landmark descriptions, more than other social media sources, which mostly contain geotagged photographs only. Wikipedia categories are mostly useful for a geospatial entity search; they offer a wide range of concepts and themes linked to the entry by the contributors [44]. There exist 113,483 different categories in the INEX of the Wikipedia XML collection, which is organized in a category graph, whereas each entry can be associated with several different categories (on average, each entry is associated with 2.28 categories).

### 2.2. The Properties of Landmark Salience

Identifying the characteristics of landmark's salience and prominence is an essential prerequisite for their use [45]. Raubal and Winter [7] qualified pedestrian landmarks using their visual attraction, such as façade, area, color, shape, and visibility, as well as semantic attraction resulting from the cultural and historical importance and structural attraction. Universal characteristics for landmark significance proposed in [46] that underlie our study, include permanence, visibility, the usefulness of the location, uniqueness, and brevity. An evaluation of these unique characteristics was done by experiments with users describing a route from origin to destination. The landmark characteristics were validated by the participants' choice of a specific landmark and a textual concept of the description. Brin and Page [47] proposed a weighting system to calculate weights for all mode landmark categories rather than specific instances of landmarks. Visual, semantic, and structural characteristics were used to give scores to the different categories, deriving an overall suitability score.

However, in the existing studies, the choice of landmarks was not conducted automatically, but based on the arbitrary user selection. In this study we propose to investigate Wikipedia for a comprehensive definition and evaluation of landmark thematic mining. In the next section we formulate an integrated ranking model for landmark mining and apply it to several cases to estimate salience of the chosen landmarks.

## 3. Methodology

To develop an integrated ranking model of Wikipedia entry salience, we apply a mixed methodology that combines (1) quantitative data retrieval with (2) qualitative weighting procedure, that includes questionnaires and user assessment of decision process. This innovative methodology contributes to a cost-effective data mining process detecting the most prominent landmarks associated with public interest.

The overall structure of the integrated ranking framework is presented in Figure 1. It is based on a set of key properties, namely:

1.  *Landmark geographic location*—data regarding the real-world geographic coordinates of the landmark entry (geotagged data).
2.  *Landmark category*—data describing the Wikipedia category list to search for categories that are frequently associated with salient and notable landmark entries.
3.  *Wikipedia page information/statistics*—numerical attribute data (page statistics) of the Wikipedia entry that points to the community interest and cultural importance of the landmark.

The queries for retrieving the aforementioned data are implemented using the MediaWiki tool (https://www.mediawiki.org/wiki/MediaWiki), a free and scalable software, and a feature-rich wiki implementation that uses Hypertext Preprocessor (PHP) programming to process data stored in the

Wikipedia database (here we analyze the English Wikipedia pages only). This tool uses MySQL, allowing one to submit complex queries to retrieve information from Wikipedia.

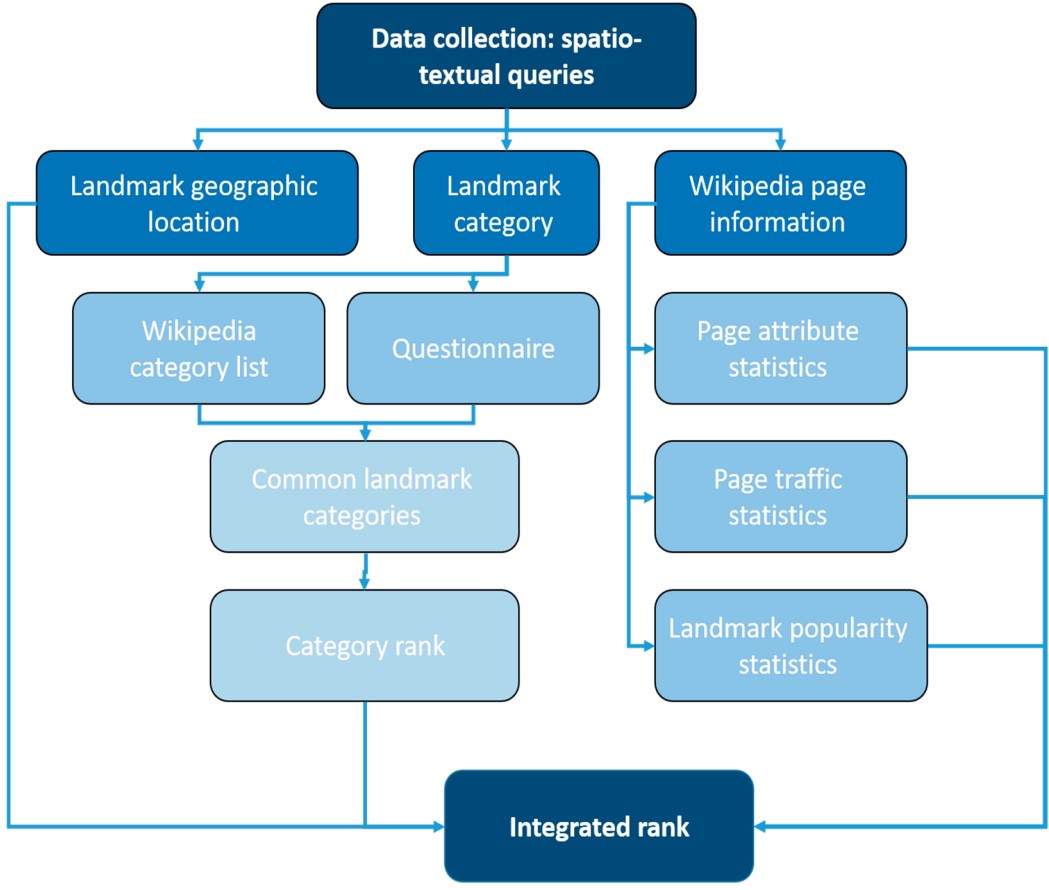

**Figure 1.** Integrated ranking model.

### 3.1. Location Properties

All Wikipedia entries describing physical objects or phenomena can store geographic coordinates regarding their location in the physical world [48]. Accordingly, many Wikipedia entries, specifically landmarks, are geotagged, storing latitude and longitude WGS84 Datum coordinates location data. Wikipedia API is used to retrieve the wiki location information. The query requires a specific location for the retrieval of the geotagged Wikipedia entries existing in the area, where normally a bounding box or a single location with defined search radius are used.

### 3.2. Category Properties

#### 3.2.1. Common Wikipedia Landmark Categories

Until 2004, Wikipedia did not have an organized system for its articles (entries), only direct links between them. From June 2004, Wikipedia has added the category pages' feature, serving as a collection of links to various articles or other category pages. Thus, it is now possible to assign articles to specific categories, and to link them to other categories, such that the category structure provides inter-links to all the Wikipedia pages based on this hierarchy.

Wikipedia relies on an enormous number of categories, which are divided into sub-categories, divided again into sub-sub-categories. Each entry has a list of relevant categories, which can be very long. Some categories do not have spatial or environmental context. Therefore, filtering at the category level allows us to retrieve the landmarks relevant to route and survey knowledge.

A preliminary query was done to retrieve all Wikipedia categories associated with physical landmarks in dense urban areas; this resulted in a list of the 50 most relevant categories. To ascertain that this list is representative, we have corroborated it with an online questionnaire (supporting information file 1). One-hundred and sixty participants from all over the world, aged 18 and up, had to choose the five most valuable landmark categories they believe are relevant to route and survey knowledge from the list. The final list was composed of 45 common landmark categories (five were not chosen by any participant), which serves for the preliminary category-based filtering (see Table 1). A Matlab code was implemented for parsing the entry's text to find and retrieve the Wikipedia entries (landmarks) associated with these categories.

### 3.2.2. Wikipedia Category Ranking

To mine the most prominent landmarks, we investigate their qualities functioning as salient and memorable in the perceptual way. The issue of landmark valuableness is subjective and could change from user to user. To resolve this, we build a hierarchical model, in which every category is ranked qualitatively and quantitatively according to its importance and valuableness. We base our model on three primary physical characteristics identified in [46], where authors aimed at differentiating between unique landmarks that are valued for route and survey knowledge. Although the authors stated five characteristics of valued landmarks, we have decided to omit at this point two of them: usefulness of location, and brevity. Both are more related to local landmarks, which are not investigated here, while for brevity authors concluded that defining it requires additional information that will increase user information processing during the navigation.

1. *Permanence*—indicates the likelihood of the landmark to be present during navigation, which can be evaluated according to temporal aspects of the physical object, e.g., how likely is that the landmark will change over or completely disappear over time (e.g., restaurant, public transportation station), or will be permanent (e.g., mountain, airport). A permanent object receives a high score, while a temporary one receives a low score. This characteristic can also differentiate between a natural and an artificial landmark, in which case a natural landmark is considered a more permanent landmark.
2. *Visibility*—indicates whether the landmark is clearly distinguished in relation to its surroundings. This characteristic illustrates general factors such as height, size, and shape on the big scale. A tall object is more noticeable from the distance, and as such it will receive a high score. In relation to size and shape, the larger and more complex the object is (in terms of area and footprint) the higher score it will gain. At this point we are occupied with overall landmark salience and consider only global landmarks, so this parameter does not necessarily reflect how the object is seen by the user. However, in the future, a more detailed spatial analysis is expected to adjust landmarks to the certain route, eyes direction, speed, transport mode, or environmental conditions.
3. *Uniqueness*—indicates the possibility that the landmark will be confused with other landmarks in the vicinity. The landmark receives a high score if it has a distinct (individual) appearance or if it is located apart from similar landmarks (e.g., park, castle), as opposed to landmarks of the same category that are more likely to be close (e.g., public transportation station, pubs).

We assign each category a score, from the less preferable (inferior) to the most preferable (superior) based on these three characteristics. To define these scores, a decision tree is implemented for each characteristic. It gives each category physical conceptions by minimizing non-numeric subjectivity in a quantitative way (Figure 2). Three decision trees are devised in the most probable score for each landmark category, with values ranging from 1 (inferior) to 5 (superior), according to the 5 leaves existing in each decision tree.

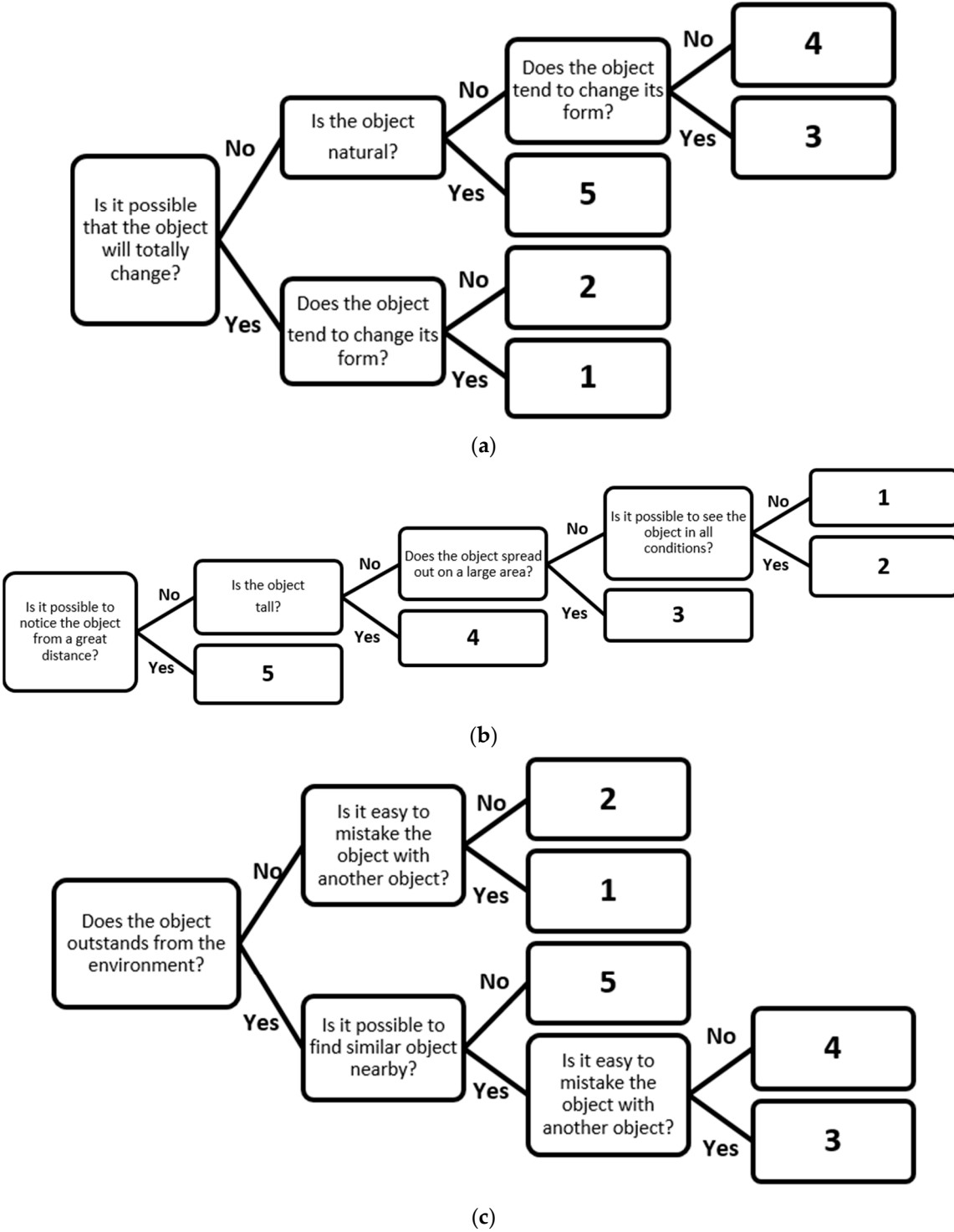

**Figure 2.** The decision trees of the three physical characteristics: (**a**) permanence; (**b**) visibility; (**c**) uniqueness.

For permanence, we examine whether it is possible that the category will change entirely and transform to another landmark; is the landmark natural or artificial; or does the landmark tend to change its form, in terms of color, shape, size, and name. For visibility, we examine whether the landmark is clearly distinguished in relation to its surroundings and if it is possible to notice it from a great distance; is the landmark tall and/or spread out over a large area; and, is it possible to see the landmark in all environmental conditions (e.g., light and weather). For uniqueness, we examine

whether the landmark is salient from its environment; is it noticeable; is it possible to find the same or similar landmark nearby; and, is it easy to confuse the landmark with other objects.

To minimize the effect of subjectivity, as well as bias resulting from local cultural and historical aspects, and to reinforce the proposed method, we approached the participants again, asking them to give scores to all 45 categories in the list according to the three decision tree classifications. Based on the scores given by 10 participants from around the world, and after validating that there were no anomalies in the values, the mean of the final category rank (CR) value is calculated for each, keeping the same weight for all three physical characteristics. Table 1 depicts the CR values for all categories, normalized between the values of 1—inferior and less valuable landmark, and 10—superior and very important and constructive landmarks for route and survey knowledge. Since numerous landmarks might exist for a specific area, the normalization of the 1–10 scale enables a richer set of rank values, and accordingly numerical classes, for the various landmarks, allowing more flexibility when choosing the landmarks.

**Table 1.** Wikipedia landmark category list with final normalized CR value for all 45 landmark categories.

| Category | Final Rank | Category | Final Rank | Category | Final Rank |
|---|---|---|---|---|---|
| Restaurant | 1 | Court | 3 | University | 7 |
| Nightclub | 1 | Market | 4 | Tall building | 7 |
| Coffee shop | 1 | School | 4 | Natural landmark | 7 |
| Pub | 1 | Museum | 4 | Cemetery | 7 |
| Bus station | 1 | Hall | 4 | Tower | 7 |
| Roundabout | 2 | Architecture structure | 5 | Hospital | 7 |
| Parking lot | 2 | Historic site | 5 | Bridge | 7 |
| Yard | 2 | Highway road | 5 | Fortress | 8 |
| Sculpture | 3 | Theatre | 5 | Airport | 8 |
| Square | 3 | Shopping centre | 6 | River | 8 |
| Landmark | 3 | College | 6 | Sky scrapper | 8 |
| Synagogue | 3 | Park | 6 | Castle | 8 |
| Library | 3 | Railway station | 6 | Mountain | 8 |
| Subway | 3 | Church | 6 | Lake | 9 |
| Hotel | 3 | Mall | 7 | Sea | 10 |

## 3.3. Wikipedia Page Properties

To prevent the retrieval of an excessive number of landmarks, together with minimizing subjectivity associated with the category ranking (CR), we have devised a supplementary ranking. This ranking is based on the Wikipedia page attributes, which are associated with the page activity. They give an indication to the community interest (popularity), cultural importance, and significance of the landmark, also in respect to salience in terms of permanence and uniqueness. As found in [47], the PageRank tool, an algorithm used by Google Search to rank web pages in their search engine results, seemed to give a wider historical and cultural perspective in weighting geographic entities. Our proposed ranking is based on the entry's numerical attributes stored on the page information (statistics). An example of these attributes and values is depicted in Figure 3.

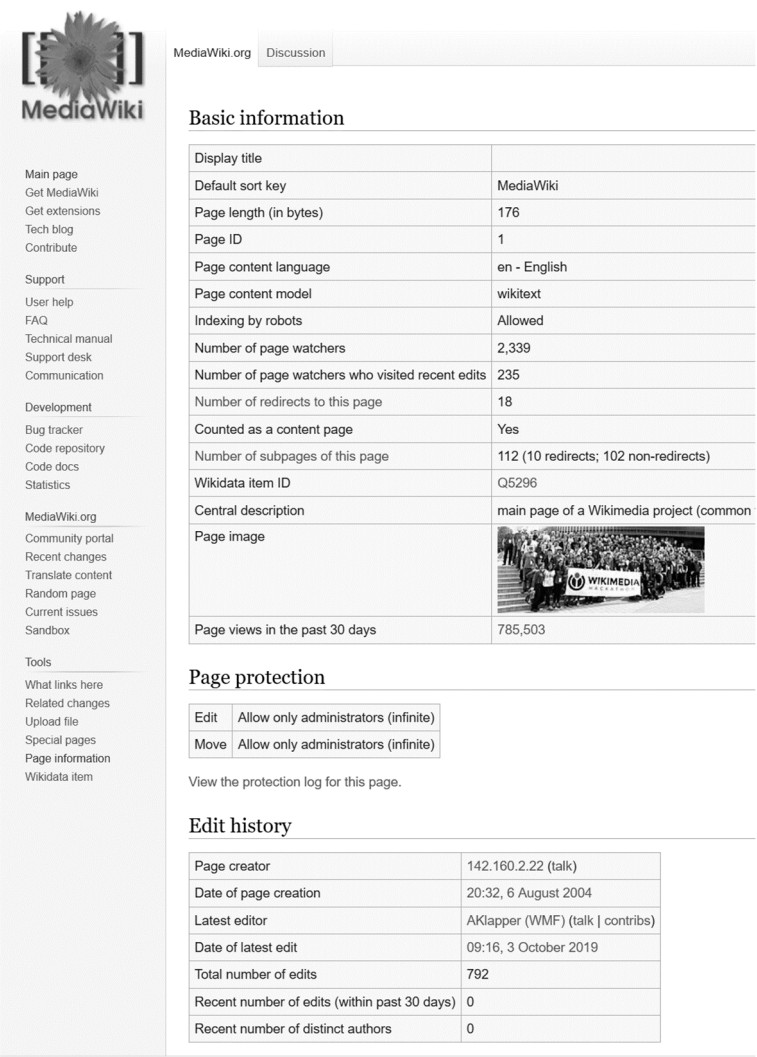

**Figure 3.** Wikipedia page statistics with attribute name (left column) and value (right column). (Source: https://www.mediawiki.org.).

Using MediaWiki API, four attributes are retrieved and analyzed for each entry, via the "Wikipedia page information":

1. *The number of redirects (NR)*—indicating the number of links that guide users from other Wikipedia pages to the analyzed article (landmark entry). A high number of redirects points to the importance and significance of the page. A prominent entry has many links and connections to other Wikipedia entries (not merely spatial ones). The frequent number of redirects is normally 1–3, thus a higher value indicates a greater importance.

2. *The date of page creation (DC)*—the date the page was created on.

3. *The date of the latest edit (DE)*—the last date the page was updated/edited on.

   Subtracting DC from DE to receive difference time (DT), we get the number of days that passed from the creation date to the recent edit date. A page created a long time ago and/or a page recently updated, will have a relatively high DT value, which indicates the relevance and importance of the page, and landmark, and its value and interest to the public.

4. *The total number of edits (TE)*—the total number of times that a page was updated/edited from the date of its creation (DC). A page that shows continuous updates suggests that new physical,

cultural or historical details are added by involved communities. Therefore, a large value of TE indicates considerable public interest in the page, and thus, in the associated landmark.

*3.4. Integrated Ranking Model*

To determine the importance of the retrieved Wikipedia landmarks, we develop a score equation, Equation (1), that weighs the main parameters: category rank (CR) from Section 3.2, TE, DT, and NR from Section 3.3. We use the least squares adjustment (LSA) technique to calculate the four unknown weight coefficients, $W_1$, $W_2$, $W_3$, and $W_4$, and we need (at least) four equations to solve the model.

$$Landmark\ Integrated\ Rank = W_1 \cdot CR + W_2 \cdot TE + W_3 \cdot DT + W_4 \cdot NR \tag{1}$$

Matrix A in the LSA relies on the data extracted from n Wikipedia articles, where i = {1,n} (i.e., *CRi, TEi, DTi, NRi*). Vector *X* is the four unknowns, i.e., weight coefficients $W_1$, $W_2$, $W_3$, and $W_4$. The main difficulty is assembling the observation vector *L*. LSA relies on the condition of minimizing the sum of squared residuals (vector V), depicted in Equation (2).

$$ATAX = ATL$$
$$X = (ATA) - 1ATL \tag{2}$$
$$V = AX - L$$

To solve this, L is compiled from two value sets: popularity statistics ($l_{popularity\_statistics}$), which gives a general idea as to how known and attractive the landmark is, and Wikipedia page traffic statistics ($l_{traffic\_statistics}$), which shows how searched/viewed the landmark is, as follows:

1.  *Popularity Statistics*—these values are retrieved from the internet website '150 most famous landmarks in the world' (http://www.listchallenges.com/150-most-famous-landmarks-in-the-world). This website gives a score for landmarks (from the top 150 landmarks around the world) according to close to 370,000 users' votes, who were asked: "How many of the 150 most famous landmarks in the world have you experienced?" The idea of using these values is of crowdsourcing, relying on the assumption that if many users have visited a specific landmark, then there must exist noteworthy values and attributes on its Wikipedia page. Forty-three landmarks are selected from this list, all having a high percentage of votes of more than 30%. The 43 landmarks generate 43 equations in the LSA model, ensuring redundancy and robustness to solve the four weight unknowns.

2.  *Traffic Statistics*—these values are retrieved from the internet website 'pageview analysis' (https://tools.wmflabs.org/pageviews/?project=en.wikipedia.org) representing the number of views of the Wikipedia article. We use the traffic statistics of 90 days to get a more comprehensive and evident perspective on the Wikipedia entry significance. The assumption is that a high value of views of a certain article gives an indication of its overall importance and interest. These values are retrieved for the same 43 landmarks used in the Popularity Statistics.

Table 2 depicts some of the values in matrices A and L used in the LSA process for the 43 landmarks. Values of L are normalized to scale all parameters between the values of 1 and 10, where $l_{popularity\_statistics}$ and $l_{traffic\_statistics}$ are averaged to form the final L vector (right-hand column). It is interesting to note that adding the supplementary Wikipedia page data (Section 3.3) to the integrated ranking model contributes to the overall assessment of valuable landmarks. For example, some famous landmarks highlighted in Table 2 that receive a CR value that is below average, for instance Buckingham Palace categorized in Wikipedia as a museum (CR of 4), Times Square categorized in Wikipedia as a square (CR value of 3), and the Statue of Liberty categorized in Wikipedia as a sculpture (CR value of 3), have relatively high page attribute and popularity values. Therefore, adding the supplementary Wikipedia page data is projected to significantly contribute to their overall high-integrated ranking.

**Table 2.** A sample of eight out of the 43 Wikipedia landmarks with parameters used in the LSA system to formulate the final integrated ranking.

| Landmark | Category | Category Rank (CR) | Number of Redirects to This Page (NR) | Latest edit-Creation Date (DT) | Number of Edits (TE) | L |
|---|---|---|---|---|---|---|
| Notre Dame | Church | 6 | 2 | 5 | 1 | 6 |
| Buckingham Palace | Museum | 4 | 2 | 9 | 5 | 7 |
| Central Park | Park | 6 | 4 | 8 | 4 | 6 |
| Empire State Building | Skyscraper | 8 | 3 | 9 | 7 | 8 |
| Times Square | Square | 3 | 3 | 8 | 3 | 8 |
| Statue of Liberty | Sculpture | 3 | 4 | 9 | 10 | 6 |
| Big Ben | Tower | 7 | 4 | 9 | 4 | 10 |
| Tower of London | Tower | 7 | 3 | 9 | 5 | 7 |

Implementing the LSA model, Equation (1) is solved as follows:

*Landmark Integrated Rank = **0.341**· CR + **0.011**· TE + **0.346**· DT + **0.268**· NR*

Weight values show that CR, DT, and NR are close to having a uniform influence on the landmark's final integrated rank, whereas TE has a very low value, and thus little influence. The Sigma a-posteriori value, calculated as $\sigma = VTV/(n - u)$, is equal to 1.5761, a value close to 1, indicating a good and balanced result of the LSA system, having a small value of the residuals.

By using the integrated ranking equation, every landmark retrieved from the Wikipedia database will have a final rank according to its category and page attributes, which values its significance.

## 4. Experimental Trials

### 4.1. Category-Based Ranking

First, we perform a search in central Berlin using the attributed location data only. The search is in the radius of 1000 meters, yielding 105 potential landmarks, i.e., landmarks having geotagged data. Figure 4 depicts the area, showing that there exist too many potential landmarks in the vicinity. This validates our statement that relying on location alone does not suffice and requires a more comprehensive classification and filtering to retrieve salient landmarks.

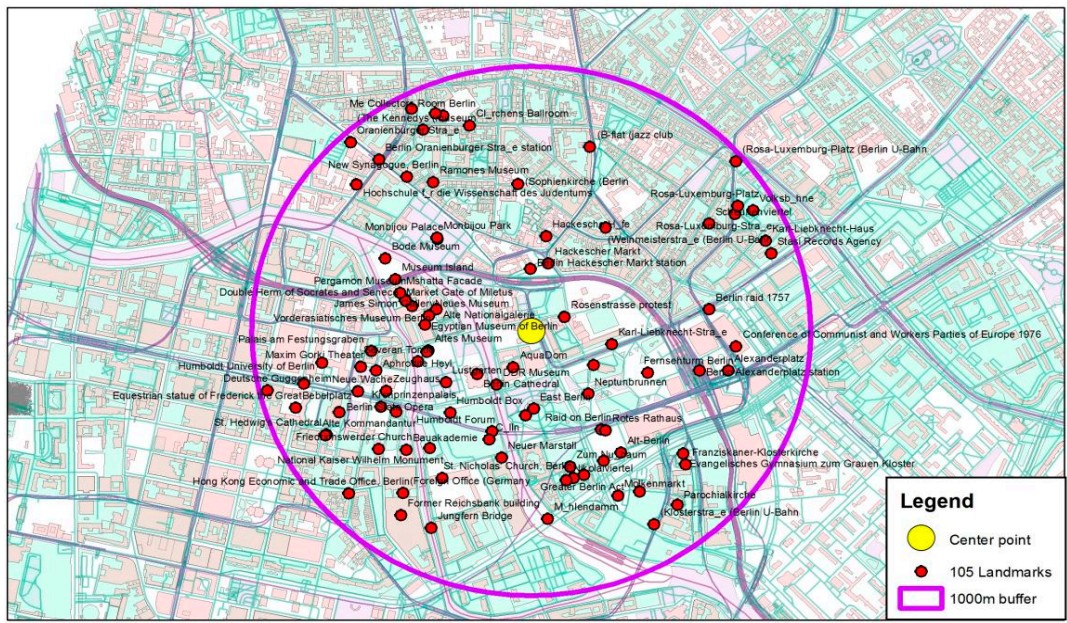

**Figure 4.** A redundant number of landmarks for a relatively small area in central Berlin (Background: © OpenStreetMap Contributors).

Each landmark from the preliminary list receives a category ranking value, according to Table 1. 47 landmarks with not listed categories are removed. The remaining 58 landmarks are divided into three classes via the Natural Breaks classification according to their category rank value (Figure 5). Those having the highest rank value of 6–7 are considered as the most salient landmarks. They consist of approximately 30% of the initial landmark list, which is still a fairly high number.

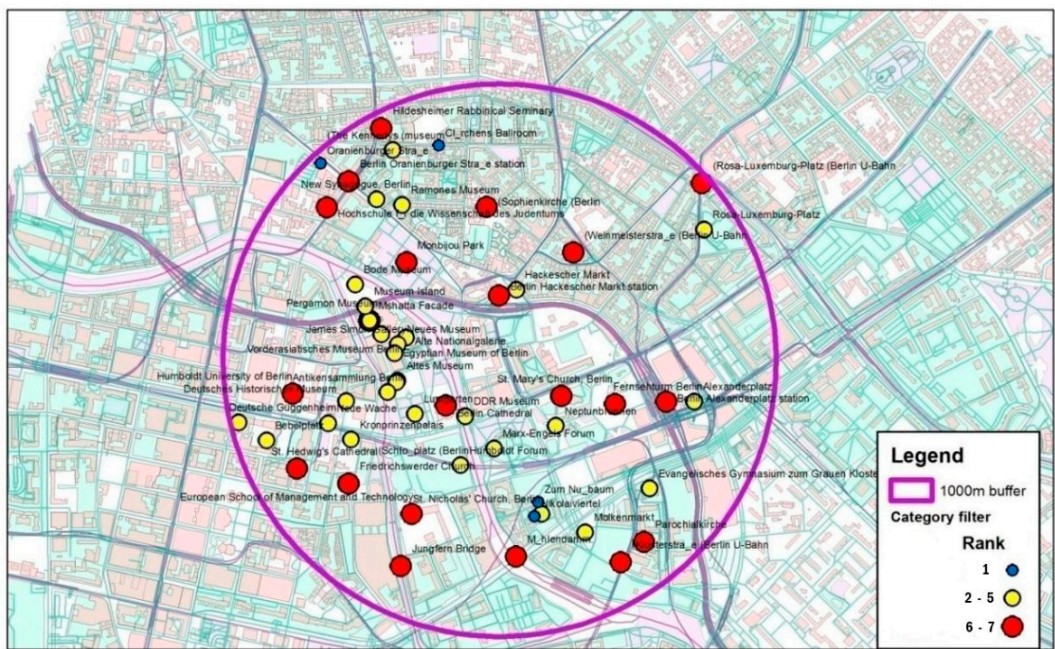

**Figure 5.** The 58 landmarks represented in different colors based on their category ranking: 1 (lowest) to 6–7 (highest). (Background: © OpenStreetMap Contributors).

Figure 6 depicts a close up of the area, where we can see that the category ranking process retrieved the following landmarks, all from the highest rank class: "Berlin Cathedral", "St. Mary's Church", and "Fernsehturm". These landmarks are noticeable and different from their close environment. Furthermore, we can see that "Zun Nu Baum", which is a restaurant and therefore is considered as a less noticeable and unique landmark, did receive a low score, and thus is not selected.

Unalike, Figure 7 depicts another zoomed area, where according to the category ranking, the high-ranking value of 7 is given to the "Hildesheimer Rabbinical Seminary", which has a "university" Wikipedia category. A medium ranking value of 5 is given to the "New Synagogue", which has a "synagogue" Wikipedia category. A visual inspection of these landmarks shows that these ranking values are not representative: the first landmark is not unique and is not prominent with respect to its close surrounding, whereas the second landmark is the exact opposite. This proves that classification based on the category ranking alone is not sensitive enough. In addition, it does not consider local cultural and historical aspects, producing biased results; a synagogue in Jerusalem is very common, mostly having an ordinary appearance just like any other building, while in Berlin it is a fairly rare unique urban facility. This example proves that ranking that relies on the category value alone does not ensure the retrieval of valuable salient landmarks.

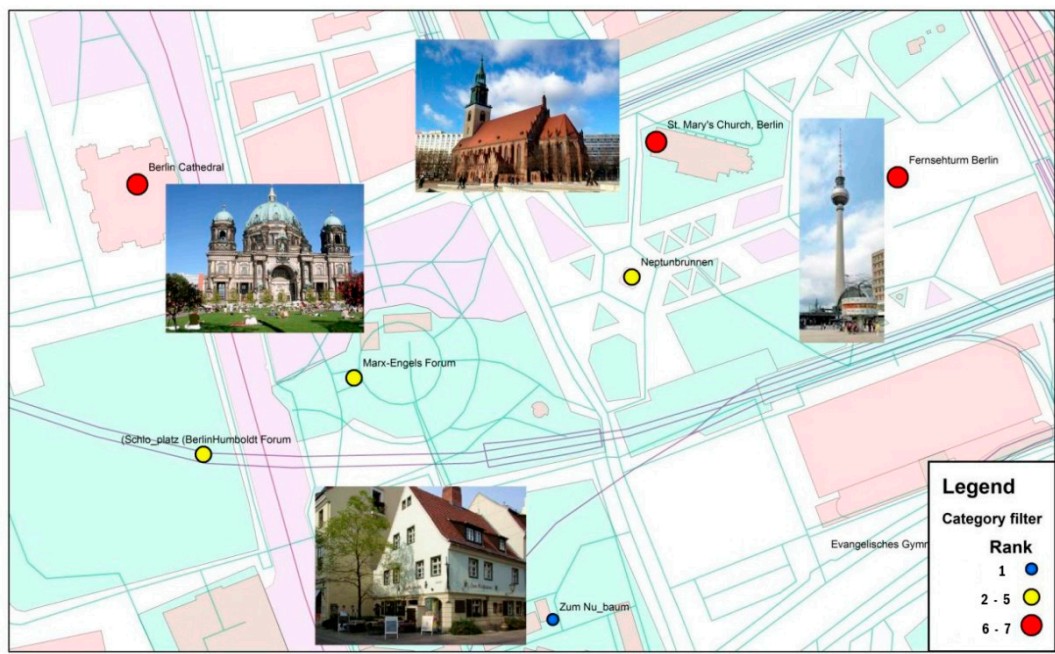

**Figure 6.** Selected area showing reliable results for the category ranking. (Background: © OpenStreetMap Contributors).

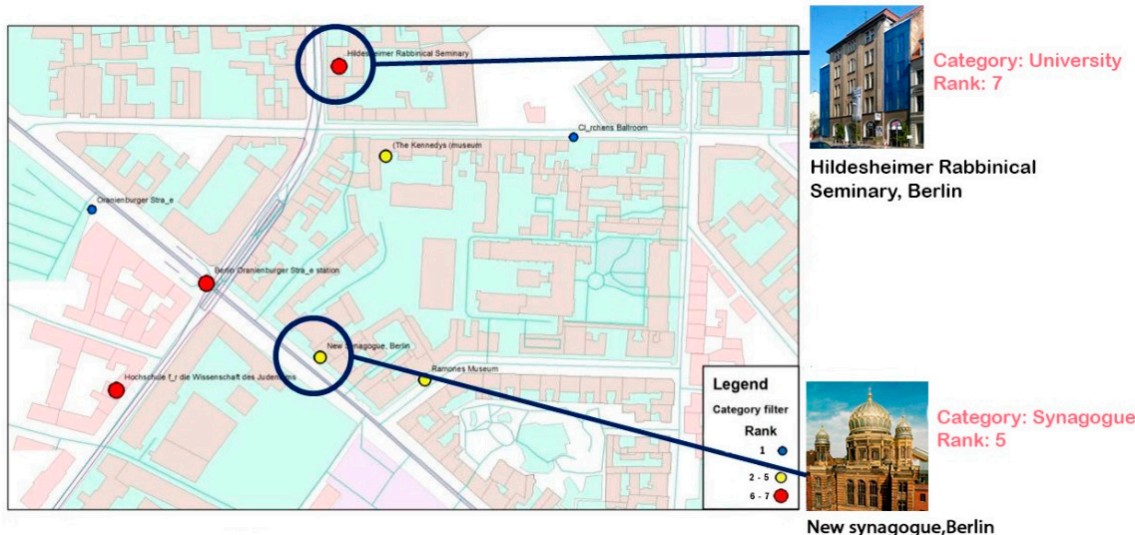

**Figure 7.** Selected area showing unreliable results of the category ranking (Background: © OpenStreetMap Contributors).

*4.2. Integrated Ranking Model*

To evaluate the robustness and effectiveness of the proposed approach, we compare the category ranking with the comprehensively integrated ranking for two examples from London. The first example, depicted in Figure 8 and Table 3, shows a low correlation, where two landmarks (top and middle rows) have CR value that is considered high (CR = 6), although both landmarks are not unique or prominent, considered as local, whereas the second is also not permanent. When the integrated ranking model is implemented, both landmarks get a relatively low value of 3, which is more representative. The opposite is presented for the third landmark (bottom), categorized as Court with CR = 3. When the Wikipedia page attributes are integrated, its final ranking value grows relatively high (6), as expected for this salient landmark.

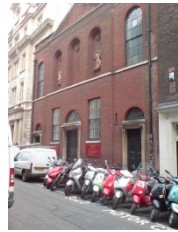 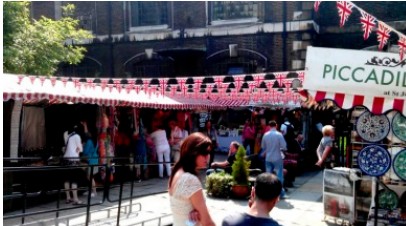 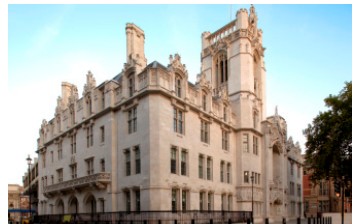

**Figure 8.** Low correlation between the category ranking (CR) value and the integrated ranking value (score). (From left to right: Church, Market, and Court. Image source: Wikipedia).

**Table 3.** Comparing ranking values that rely on category ranking (CR) and integrated (Rank).

| Landmar.k | CR | NR | DT | TE | Rank |
|---|---|---|---|---|---|
| Church of Our Lady of the Assumption and St Gregory | 6 | 1 | 2 | 1 | 3 |
| Piccadilly Market | 6 | 1 | 2 | 1 | 3 |
| Supreme Court of the United Kingdom | 3 | 6 | 8 | 5 | 6 |

Figure 9 and Table 4 depict an example where a high correlation exists between the values, thus the integrated ranking model ascertains the landmark significance and salience. A Tower, Church, and River, all having high category ranking values (CR = 6 − 8). When the Wikipedia page attributes are used, their final integrated ranking value remains high, justified by their character and saliency—all are prominent and unique urban features in their surroundings, serving as global landmarks.

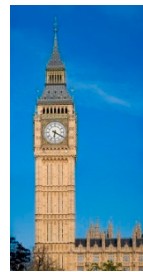 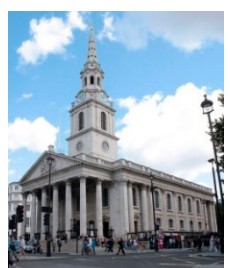 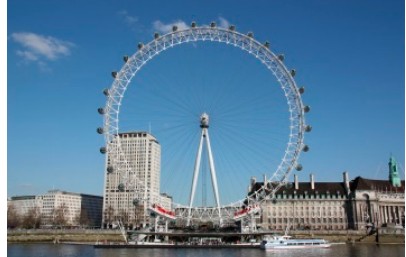

**Figure 9.** High correlation between the category ranking (CR) value and the integrated ranking value (score). (From left to right: Tower, Church, and River. Image source: Wikipedia).

**Table 4.** Comparing ranking values that rely on category ranking (CR) and integrated (Rank).

| Landmark | CR | NR | DT | TE | Rank |
|---|---|---|---|---|---|
| Big Ben | 7 | 7 | 10 | 10 | 8 |
| St Martin-in-the-Fields | 6 | 10 | 9 | 2 | 8 |
| London Eye | 8 | 6 | 10 | 10 | 8 |

*4.3. Comparative Evaluation*

This section finalizes our trials by evaluating the comprehensiveness of the proposed integrated ranking model. To achieve that, we conduct a comparative analysis of two cities with different urban scale and cultural character—New York and Tel Aviv. These cities have diverse urban density and morphology that influence the availability and heterogeneity of their landmarks. Hypothetical touristic routes are chosen to examine the practicality and robustness of the research idea, presumably filtering "less salient" landmarks, and remaining with only several salient ones.

### 4.3.1. New York

In the New York route, which is 2.2 km long, 463 landmarks are retrieved using the location query, resulting in 269 landmarks with the signified category. Since New York is a very dense urban environment, with many high-rise buildings, we use a 100 m buffer around the route to have a closer analysis of the landmarks.

A preliminary examination shows that the less salient landmarks, such as hotels and restaurants, receive low rankings. Several theaters and elevated transit lines, for example, which have a relatively high rank, receive a low integrated rank due to the use of the supplementary Wikipedia page parameters. The output is divided into 3 classes via the Natural Breaks classification, where the red color represents the group with the highest ranking value. Figure 10 depicts the selected route with the landmarks retrieved and their final ranking values. Thus, the preliminary 13 landmarks having high category value (6 and up) are reduced to 8 by using the integrated ranking model.

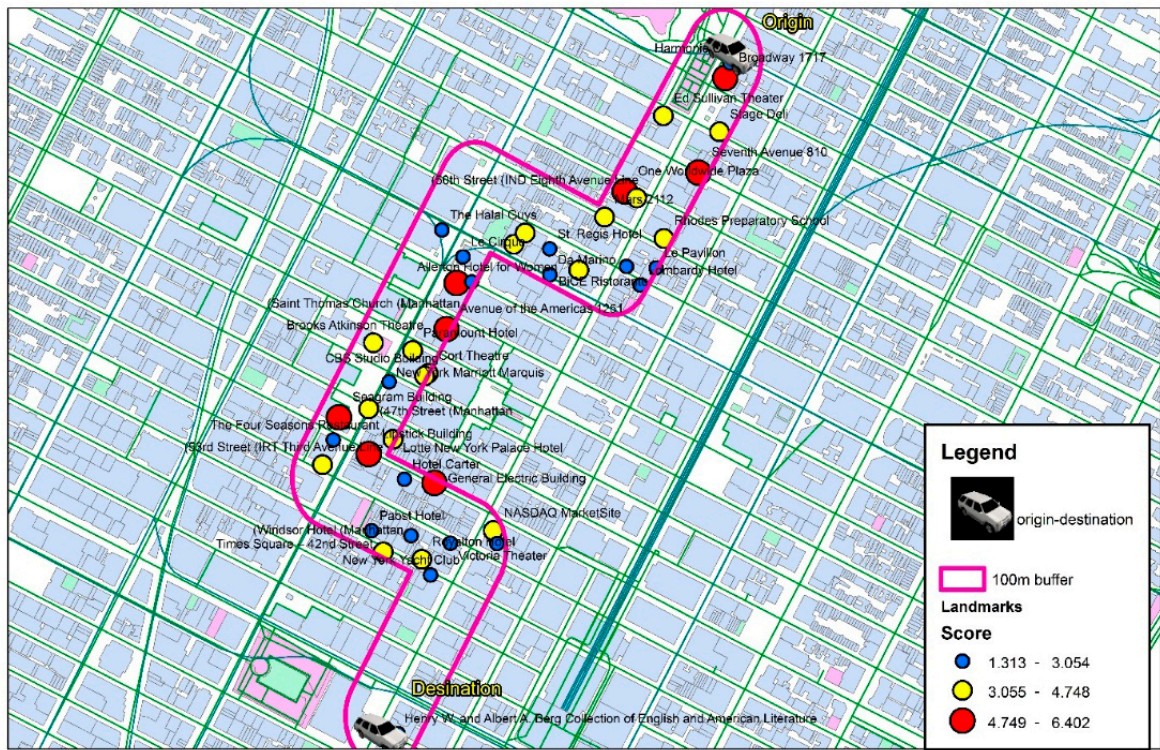

**Figure 10.** New York route superimposed with landmarks. Colors depict score value classes according to the integrated ranking model. (Background: © OpenStreetMap Contributors).

The eight remaining landmarks are an adequate number to communicate during navigation in a route of this length [13,49]. Table 5 summarizes the landmarks' classification and filtering statistics of the overall process.

**Table 5.** Statistics summary of the Wikipedia landmark integrated ranking model for the New York route.

| Selection by | Number of Landmarks | Number of Landmarks after Filtering | % Filtered |
| --- | --- | --- | --- |
| Common category list | 463 | 269 | 42 |
| Buffer | 269 | 45 | 90 |
| Category rank value | 45 | 13 | 97 |
| Integrated (score) value | 13 | 8 | 98 |

Inspecting the resulting eight landmarks, we validate that they all are very distinctive and important environmental features: towers and skyscrapers. Although they represent a somewhat homogenous landmark group, this is not a surprising result for a big city, such as New York. This is not a final, optimal solution that could be used for a navigation. Additional adjustment will be needed for each of the travel mode, where this one is probably the most closest for pedestrians. Depending on the case, a more heterogeneous array of landmarks should be communicated to ensure the better spatial orientation and environmental perception, one that considers other salient landmarks with the medium integrated ranking score.

The aforementioned results rely on a global retrieval process. However, by locally dividing the route into segments using critical decision points (mainly route turns), one can extract more relevant landmarks. When integrated into the navigation process, landmarks situated near route-turns will contribute the most, and thus should be prioritized. Based on a local process, the selection of more tuned landmarks becomes possible, although the landmarks might have lower integrated ranking score values. An example is depicted in Figure 11a, where relying on a local Natural Breaks classification the "Museum of modern art" is not selected since it has a medium score value for that segment. At the next segment of the route, depicted in Figure 11b, the same landmark is selected having a high score value for that segment. This proves that local analysis derived from the geometry of the route should be considered to produce route-related landmarks. Such an analysis considers the 'most' salient landmarks, i.e., heterogeneous ones nearby critical decision points, while still validating that not too many landmarks are communicated.

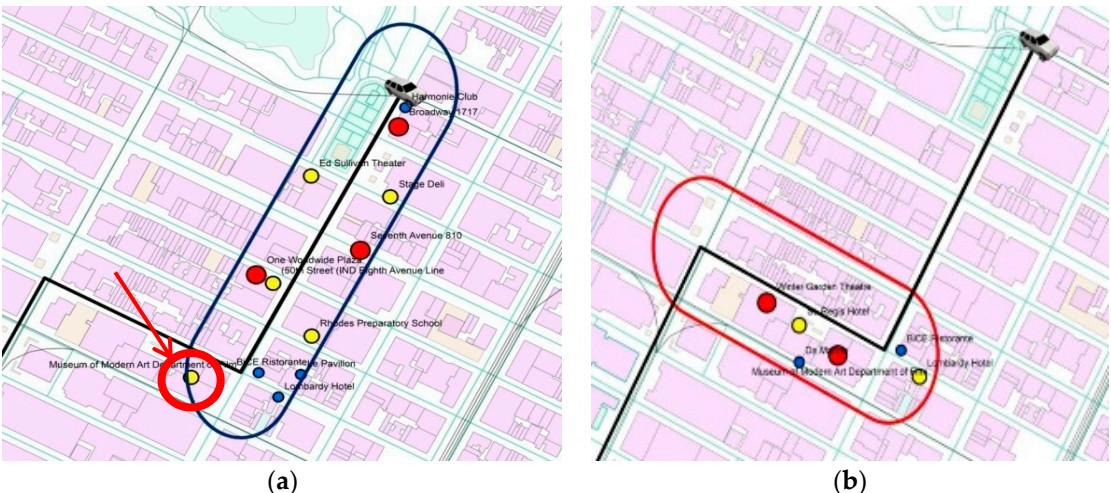

**Figure 11.** Local classification per segment showing landmarks that are filtered (**a**) and considered (**b**) based in their score in relation to other landmarks in the segment. (Background: © OpenStreetMap Contributors).

### 4.3.2. Tel Aviv

Our next evaluation is of a route in central Tel Aviv, 1.7 km in length. Only seven landmarks are retrieved—much less comparing to New York. This can be explained due to the fact that we rely on the English Wikipedia pages, and not the Hebrew ones (if we apply similar queries on the Hebrew Wikipedia we receive several dozens of pages having the desired landmark categories). Considering the group with the highest score, we remain with only one valuable landmark for the entire route. However, by increasing the buffer radius to 200 meters, four additional high score value landmarks are added (Figure 12). This is a better result—the landmarks retrieved for this route are valuable and informative. In the case of Tel Aviv, we see a much more heterogeneous group of landmarks comparing to New York, e.g., skyscrapers, museum, and a shopping mall. A garden, a library, and a shopping center—all are less salient landmarks—receive lower ranking. Another interesting result

is the capacity of the model to prioritize landmarks situated close-by. As shown in the lower right corner of Figure 12, a skyscraper and a tower that share the same location were selected; however, the developed integrated ranking model automatically chooses the skyscraper as having a higher rank, which is the optimal solution here.

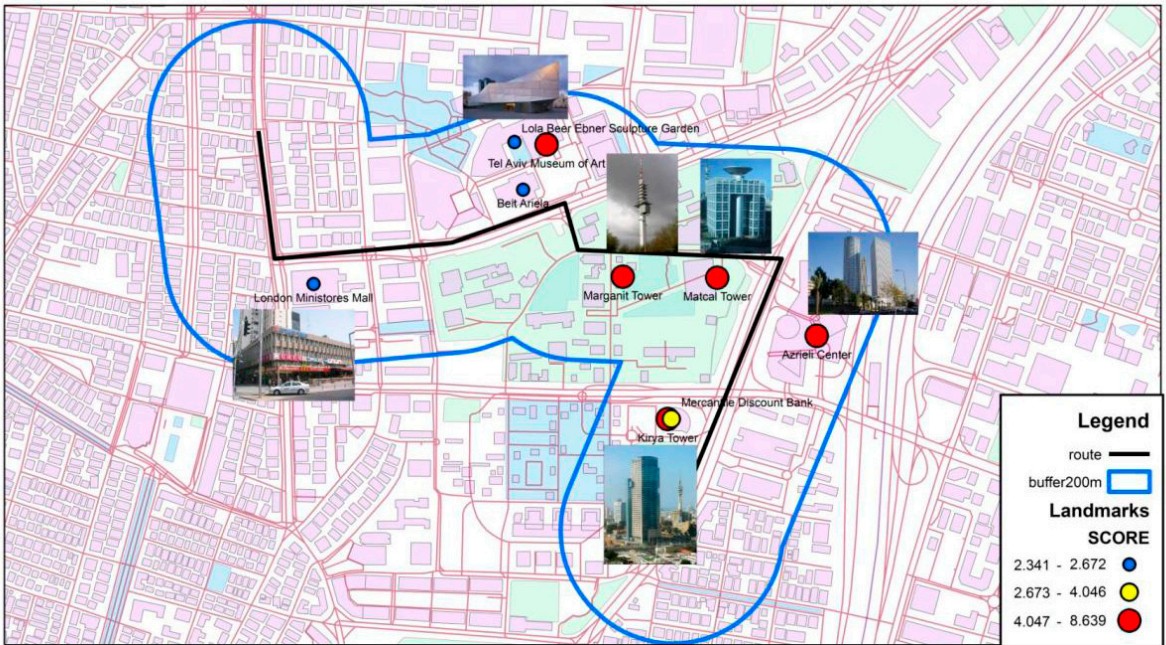

**Figure 12.** Tel Aviv route with 200-meter buffer superimposed with landmarks. Colors depict score value classes according to the integrated ranking model. (Background: © OpenStreetMap Contributors).

## 5. Discussion and Future Work

This research presents a methodology to utilize user-generated public-domain geotagged content for retrieving salient landmarks, showcasing Wikipedia. We focus here on the development of the data mining and filtering algorithms for landmark salience weighting. These rely on a qualitative set of cognitive measures integrated with quantitative parameters associated with public interest.

First, the category ranking alone proved to be to some extent robust, having problems that result from its subjective definition, disregarding cultural and historical aspects, and hence producing biased results. To overcome this, a complementary and more comprehensive integrated ranking model is suggested, based on a list of attribute values associated with the retrieved Wikipedia page properties. The outcomes of the presented trials and evaluations demonstrate the retrieval of the significant and prominent landmark entries. In addition, our results show that user-generated data and information sources can serve as a working ground for the retrieval of landmarks, and be used further in navigation systems.

Future work is planned on the integration of local versus global classification, and further adjustment to diverse navigation modes. Applied here for detection of general salience, we strive to see whether and how the model could be transferable for navigational aid at different travel modes and different routes. Landmark extraction for car navigation requires more distinct landmarks and less localized ones, which can be fundamental for pedestrians. However, we see no limitations for adjusting the methods to match different modes of travel. For example, the retrieval of landmarks per route or per segment; a local classification that takes into account the geometry of the route (i.e., decision points) will demonstrate more tuned results.

We also suggest further development of the geospatial attribution, such as evaluating of landmark visibility along the route. To introduce a more sophisticated method of visibility analysis, first the travel mode should be identified together with corresponded decision points and typical viewing distance.

Adding visibility constraints with respect to urban or rural morphologies, such as line-of-sight and viewshed in 3D environments, can also contribute to the overall process by communicating landmarks that are visible to users.

Moreover, along the route at decision points, it would be helpful to distinguish different geometries of landmarks—point-like, linear, etc. The size and shape of the landmark bear useful information for wayfinding instructions [50]. Additionally, adjustments could be done to match a local culture, urban density, and scale. Current study analyses the English Wikipedia pages only, whilst we believe that analyzing Wikipedia pages in other languages will contribute to retrieving more landmarks, as well as extracting more relevant local knowledge.

Routes in rural areas have to be analyzed as well, to assess the algorithms in areas that might show an insufficient number of noticeable landmarks. Additionally, issues related to the homogeneity of landmarks around the route need to be investigated, developing the ability to ensure the retrieval of a heterogeneous set of landmarks, which would better contribute to route and survey knowledge.

This research establishes a new knowledge discovery related to landmarks retrieval from UGC, overcoming limitations that are commonly associated with authoritative databases. The developed GIR integrated ranking model contributes to the understating and implementation of data mining methods dealing with big data. It will allow more detailed implementation addressing navigation and wayfinding, assisting users to create constructive mental maps, contributing to better spatial orientation, awareness, and holistic environmental perception.

**Supplementary Materials:** The following are available online at http://www.mdpi.com/2220-9964/8/12/529/s1, Supplementary Materials File 1: Online questionnaire - Landmark Categories

**Author Contributions:** Conceptualization, Noa Binski and Sagi Dalyot; data curation, Sagi Dalyot; formal analysis, Asya Natapov and Sagi Dalyot; funding acquisition, Asya Natapov and Sagi Dalyot; investigation, Asya Natapov and Sagi Dalyot; methodology, Noa Binski and Sagi Dalyot; resources, Sagi Dalyot; software, Noa Binski; supervision, Sagi Dalyot; validation, Noa Binski and Sagi Dalyot; visualization, Noa Binski and Asya Natapov; writing—original draft, Noa Binski, Asya Natapov, and Sagi Dalyot; writing—review and editing, Asya Natapov and Sagi Dalyot.

**Funding:** This research was partially funded by the EUROPEAN UNION'S HORIZON 2020 research and innovation programme under the Marie Skłodowska-Curie Actions [MSCA IF grant number 744835].

**Conflicts of Interest:** The authors declare no conflict of interest. The funders had no role in the design of the study; in the collection, analyses, or interpretation of data; in the writing of the manuscript, or in the decision to publish the results.

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
