# Peer review of "Retrieving Landmark Salience Based on Wikipedia: An Integrated Ranking Model"

_ijgi, doi:10.3390/ijgi8120529_

Round 1
Reviewer 1 Report
I have no comments apart from that some fonts in Figures 2,3,4,5,6,7,10,11,12 are quite small and often unreadable. I recommend removing the text that is not needed and enlarging that that is required.
Author Response
|
Reviewer #1 |
|
I have no comments apart from that some fonts in Figures 2,3,4,5,6,7,10,11,12 are quite small and often unreadable. I recommend removing the text that is not needed and enlarging that that is required. |
|
We would like to thank the reviewer for acknowledging potential of our study. According to this comment we enlarged the above figure fonts where possible. |
Reviewer 2 Report
This submission is a very interesting paper with a very applied topic and falls in the scope of the IJGI journal. Although, it has some potential for improvement.
Introduction
The general description of the problem (Introduction) and the description of its importance for the science and the society could be further improved. The degree of innovativeness of the methodological approach is not convincingly demonstrated. Some more details about its innovative features could further improve the quality of this paper. Why is this paper likely to be cited in the future?Method
A bit more text regarding the originality of this work and why it contains new results that significantly advance the research field.Results
I believe that adding a bit more text on why the results of the method are satisfactory (evaluation approach) will increase the quality of this work Could the results be more satisfactory if you have changed something in the methodology? Are the results (or the method) sensitive to this specific study area?Discussion
In the Discussion section I would have wished to see more information on the actual meaning of the findings and how the results add to the broader topic as well as the specific scientific fieldConclusion
The "Conclusions" section, could be further improved by describing the importance of this work, the highlight of potential further development of this methodology.Author Response
|
Reviewer #2 |
|
This submission is a very interesting paper with a very applied topic and falls in the scope of the IJGI journal. Although, it has some potential for improvement. |
|
Thank you for acknowledging potential and relevance of this study. |
|
The general description of the problem (Introduction) and the description of its importance for the science and the society could be further improved. The degree of innovativeness of the methodological approach is not convincingly demonstrated. Some more details about its innovative features could further improve the quality of this paper. Why is this paper likely to be cited in the future?
Method A bit more text regarding the originality of this work and why it contains new results that significantly advance the research field.
|
|
Following this useful comment, we elaborated more on the innovative aspect of the study, its originality, and significance of the results that advance the research field. Please see P.2-3, L. 89-100; P.3.L. 91-96, P.3.L.110-114, P.5, L. 170-173. |
|
Results I believe that adding a bit more text on why the results of the method are satisfactory (evaluation approach) will increase the quality of this work. Could the results be more satisfactory if you have changed something in the methodology? Are the results (or the method) sensitive to this specific study area? |
|
We used here a gradual and comparative evaluation approach - first we examine a category ranking, then integrate it with a supplementary ranking based on the Wikipedia page attributes associated with the page activity. The results are satisfactory as shown in section 4.3 (P.16, L.468-472). They are also sensitive to specific area as shown on P.14. L.424-431 and P.19, L.529-530. |
|
Discussion In the Discussion section I would have wished to see more information on the actual meaning of the findings and how the results add to the broader topic as well as the specific scientific field. The "Conclusions" section, could be further improved by describing the importance of this work, the highlight of potential further development of this methodology. |
|
Following this comment we elaborated more on the actual meaning of the findings and their significance for the broader area of the research. Please see P.20, L.604-609. |
Reviewer 3 Report
The classification method is interesting, but like all methods based on a set of heuristics, it involves biases, i.e. if the assumptions on which it is based are intuitively valid, they are not systematically so. The experiments provided give good indications of the effects of the method. A precise assessment of these limits would require many much broader and more important tests, but would probably be the subject of future work. However, it seems to me necessary that a thorough discussion of these invalid situations be made and that their impact be estimated.
The usefulness of the visibility criterion seems questionable to me. As the authors note, this criterion does not guarantee that the driver will be able to see the site, particularly when it is needed. It seems to me that a more sophisticated mechanism is essential to be really useful. Moreover, is a site that can be seen from very far away really a useful benchmark to guide you?
The criterion of uniqueness also seems to me to be more sensitive than estimated by the authors. There are certain types of sites, particularly in tourist areas, such as restaurants or bars, that could be used as landmarks if their names are specified, for example.
The process described in Figure 11 is not very explicit: are additional benchmarks detected because they increase the value of their rank (and in this case, it should be explained why and how) or because their value is among the most important values for this road segment?
On the form, the state of the art includes many sentences built on the model:"[reference] does this", which does not make reading very easy and also reduces the state of the art to a list of short descriptions of projects whose links and overall impact are not easy for the reader to understand.
Author Response
|
Reviewer #3 |
|
The classification method is interesting, but like all methods based on a set of heuristics, it involves biases, i.e. if the assumptions on which it is based are intuitively valid, they are not systematically so. The experiments provided give good indications of the effects of the method. A precise assessment of these limits would require many much broader and more important tests, but would probably be the subject of future work. However, it seems to me necessary that a thorough discussion of these invalid situations be made and that their impact be estimated. |
|
We agree with the reviewer that methods proposed in this paper involve some biases. The inclusion of landmarks into navigation systems is a long-standing goal. First, the system has to be able to extract suitable points of interest and to assess their salience in the role of potential landmarks. Then, they have to be integrated in meaningful ways adjusting to the particular travel mode, navigation context or appropriate arrangements. Here we focus only on the first stage, conducting landmark retrieval and estimating landmark salience. This stage is paving the way for the next one that requires an alternative set of preconditions to define an appropriate landmark for a particular wayfinding task. (Please see P.2, L63-88). In addition, following this comment, we discuss a precise assessment of the study limits that could be addressed in a broader test. Please see future work paragraph on P.18, L.499-506. |
|
The usefulness of the visibility criterion seems questionable to me. As the authors note, this criterion does not guarantee that the driver will be able to see the site, particularly when it is needed. It seems to me that a more sophisticated mechanism is essential to be really useful. Moreover, is a site that can be seen from very far away really a useful benchmark to guide you? |
|
In this study we adopt Burnett’s et al. (2001) characteristic of ‘visibility’, i.e., whether the landmark is clearly noticeable in relation to its surroundings and the environmental conditions (P.6, L232-240). As noticed by the reviewer, we refer to the prominence and distinction of the landmark, and not to whether the landmark is seen by the user (this requires a complex 3D analysis, the use of comprehensive 3D city models and prior knowledge of the navigation context). At this point, our study is focused more on data-mining and data-interpretation in terms of geometric definitions. Therefore, it pays less attention to geospatial attribution and spatial analysis, so the significance of landmark visibility perceived by users is not addressed. However, these stages are suggested and detailed in the Discussion and Future Work chapter (P.19 L. 510-515). |
|
The criterion of uniqueness also seems to me to be more sensitive than estimated by the authors. There are certain types of sites, particularly in tourist areas, such as restaurants or bars, that could be used as landmarks if their names are specified, for example. |
|
We agree with the reviewer that characteristic of ‘uniqueness’ could have much wider interpretation. Here we limited ourselves to Burnett et al. (2001) where uniqueness indicates the possibility that the landmark will be confused with other landmarks in the vicinity, i.e., how common is a specific landmark category to a specific area, and also what are the chances that two (or more) similar adjacent landmarks of the same category exist. On P.13, L.383-392 we discuss that this characteristic does not consider local uniqueness and historical aspects. However, in addition to the uniqueness criterion, we use a score derived from a set of numerical attributes that are associated with public interest in the Wikipedia page – these include number of redirects and the date of the latest edit. Due to that a rating of famous restaurants or bars in touristic areas should be improved. It is important to note that on the current stage we only extract suitable points of interest and assess their salience in a role of potential landmarks. In the further research these will be integrated in navigation systems adjusting to the particular travel mode, context or urban arrangements. Please see for details P.18-19, L. 502-509 and P.19, L. 516-526. |
|
The process described in Figure 11 is not very explicit: are additional benchmarks detected because they increase the value of their rank (and in this case, it should be explained why and how) or because their value is among the most important values for this road segment? |
|
The additional landmark in Figure 11 is selected because it increased the value due to dividing the route into segments. The performed test demonstrate that more relevant landmarks can be extracted by adjusting to a current road segment. in Figure 11b, the same landmark that was filtered in Figure 11a, is now selected because it has a higher score value for that particular segment. |
|
On the form, the state of the art includes many sentences built on the model:"[reference] does this", which does not make reading very easy and also reduces the state of the art to a list of short descriptions of projects whose links and overall impact are not easy for the reader to understand. |
|
The referencing style is a prerequisite of the journal; however, we rewrote the state of the art paragraph to make reading easier and to link together references’ overall impact on the theme. Please see P.2, L.63-88. |
